# Phylogeography of 27,000 SARS-CoV-2 Genomes: Europe as the Major Source of the COVID-19 Pandemic

**DOI:** 10.3390/microorganisms8111678

**Published:** 2020-10-29

**Authors:** Teresa Rito, Martin B. Richards, Maria Pala, Margarida Correia-Neves, Pedro A. Soares

**Affiliations:** 1Life and Health Sciences Research Institute (ICVS), School of Medicine, University of Minho, 4710-057 Braga, Portugal; teresarito@med.uminho.pt (T.R.); mcorreianeves@med.uminho.pt (M.C.-N.); 2ICVS/3B’s, PT Government Associate Laboratory, University of Minho, 4710-057 Braga, Portugal; 3Department of Biological and Geographical Sciences, School of Applied Sciences, University of Huddersfield, Huddersfield HD1 3DH, UK; m.b.richards@hud.ac.uk (M.B.R.); M.Pala@hud.ac.uk (M.P.); 4Centre of Molecular and Environmental Biology (CBMA), Department of Biology, University of Minho, 4710-057 Braga, Portugal; 5Institute of Science and Innovation for Bio-Sustainability (IB-S), University of Minho, 4710-057 Braga, Portugal

**Keywords:** phylogeography, phylogenomics, molecular epidemiology, travel bans, intercontinental founders, SARS-COV-2 lineages

## Abstract

The novel coronavirus SARS-CoV-2 emerged from a zoonotic transmission in China towards the end of 2019, rapidly leading to a global pandemic on a scale not seen for a century. In order to cast fresh light on the spread of the virus and on the effectiveness of the containment measures adopted globally, we used 26,869 SARS-CoV-2 genomes to build a phylogeny with 20,247 mutation events and adopted a phylogeographic approach. We confirmed that the phylogeny pinpoints China as the origin of the pandemic with major founders worldwide, mainly during January 2020. However, a single specific East Asian founder underwent massive radiation in Europe and became the main actor of the subsequent spread worldwide during March 2020. This lineage accounts for the great majority of cases detected globally and even spread back to the source in East Asia. Despite an East Asian source, therefore, the global pandemic was mainly fueled by its expansion across and out of Europe. It seems likely that travel bans established throughout the world in the second half of March helped to decrease the number of intercontinental exchanges, particularly from mainland China, but were less effective between Europe and North America where exchanges in both directions are visible up to April, long after bans were imposed.

## 1. Introduction

Towards the end of 2019, a novel virus jumped species from an unidentified host to humans, associated with an outbreak of pneumonia [1]. This is thought most likely to have taken place in the Huanan live animal and seafood “wet market” of Wuhan, China, although problems with this scenario have fueled alternatives [2]. This virus, identified as a member of the coronavirus family, showed evolutionary proximity to viruses found in bats [3,4], a known natural reservoir of coronaviruses [5,6,7], and also pangolins [8]. The novel virus, named SARS-CoV-2, is a positive single-stranded RNA beta-coronavirus with a genome 29,903 nucleotides long [9].

SARS-CoV-2 causes severe acute respiratory disease in humans, known as coronavirus disease 2019 (COVID-19), which may also include gastrointestinal and neurological symptoms [10,11,12,13,14]. The virus has spread dramatically [10]; in three to four months, it was officially detected in at least 200 countries, causing a global pandemic with around 22 million infected individuals and more than 1,000,000 deaths (as of 29/09/20; source: World Health Organization), a scale not seen in the last 100 years.

Traditional evolutionary approaches have been applied to study the phylogeny of SARS-CoV-2. These are, in particular, maintained and updated on the GISAID website [15]. However, the similarity of variation in SARS-CoV-2 to variation in the human mitochondrial DNA (mtDNA) genome has also been noted on several occasions, leading to several attempts to use the tools of human mitochondrial intraspecific phylogeography to track the spread of the virus. The first of these, published in April 2020, used just 160 genomes [16], whilst another, for which a preprint was posted in June 2020 [17], and subsequently published [18], included ~4700, albeit focusing on ~3400 higher-quality sequences.

The first study [16] was undeniably pioneering, especially in its application of phylogeographic principles [19] to the spread of the pandemic but received criticism for relying upon a widely used but very weak network reconstruction algorithm, as well as the small dataset [20,21]. The median-joining (MJ) algorithm [22]—a fast, heuristic approach designed primarily to graphically display the variation in hypervariable human microsatellite data—is poorly suited to the task of phylogenetic reconstruction, and fails especially to correctly reconstruct long branches that particularly affect the identification of the root [21,23]. Although a more precise phylogenetic reconstruction is considerably more challenging and labor-intensive, MJ does not provide a reliable short-cut, and at minimum should be combined with a pre-processing step using the reduced-median network algorithm, as recommended by its originators [22]. Other studies currently available only as preprints have also adopted the MJ approach, e.g., Song et al. [24].

A more recent study [18] applies a more sophisticated approach to a much larger dataset, using both maximum parsimony and maximum likelihood, including a thorough analysis of rate variation across the genome. They pinpointed the arrival of the virus in its first human host with striking precision to November 12th, 2019. Nevertheless, this study still struggled to identify the root of the phylogeny and focused primarily on the specific issue of the impact of super-spreaders [25] during the course of the pandemic.

Here, we analyzed almost an order of magnitude more sequences than this most recent study and applied novel methodologies to establish a more confident identification of the root. Like the previous studies referred to above, we co-opted approaches from human intraspecific phylogeography, as applied in the past, especially to mitochondrial DNA. In particular, we used the reduced-median (RM) network algorithm [26] to generate the phylogeny. The RM algorithm explicitly reconstructs ancestral nodes in the phylogeny as median vectors, of which MJ only achieves a small subset [22,26]. The RM algorithm was explicitly developed for intraspecific human mtDNA variation [26] and as such is highly appropriate because the variation in SARS-CoV-2 genomes is strikingly similar to that in whole mitochondrial genomes. The number of viral genomes that we analyzed (~27,800) was similar to the publicly available mitochondrial genome database (PhyloTree) [27].

In our study, we highlighted a major feature of the spread of the virus to which phylogeographic analysis was uniquely suited and to which the study of mitochondrial genomes has made important contributions in the past. This is the extent and impact of founder effects on the dispersal of the virus around the human populations of the world. Furthermore, highlighting intercontinental dispersals of the various virus lineages allowed us to comment on the effectiveness of the early travel restrictions implemented by some countries.

## 2. Materials and Methods

We collected SARS-CoV-2 complete genomes from the GISAID website [15] on 14 May 2020. We excluded only genomes labelled as low coverage. We obtained a total of 27,812 sequences (Appendix A). Additionally, we used three sequences as outgroups: two from viruses present in bats that showed proximity to SARS-CoV-2 (obtained from the GISAID website, RmYN02/2019|EPI_ISL_412977, and BetaCoV/bat/Yunnan/RaTG13/2013 (GWHABKP00000001)) and one from a virus present in pangolin from GenBank (MT084071).

We aligned sequences using the MAFFT version 7 online [28]. The beginning and end of many sequences showed low-quality. Following the alignment, we therefore excluded the first and last 100 bps from all sequences, as well as sequences that showed more than 5% of missing data in the alignment due to low-quality. We compared the remaining sequences with the reference sequence (NC_045512) and annotated variants using mtDNAGeneSyn [29]. This software lists all variants as the position in the reference sequence only (in case of transitions) or the position with the derived nucleotide (for transversions). The software also registers gaps in the sequences. Some sequences displayed a high number of small regions with low-quality (gaps between 10 and 200 bps). As these numerous gaps indicated a general low-quality of the sequences, we excluded those that showed over 20 of those gaps.

We used the reduced-median algorithm [26], implemented in the Network 10 software, to construct reduced-median networks of the data. Unlike the median-joining algorithm [22] recently used to analyse a small SARS-CoV-2 dataset [16], the reduced-median algorithm is an evolutionary network-building methodology specifically designed for use with mitochondrial DNA, which the SARS-CoV-2 variation closely resembles. Given the fact that the dataset, even considering the quality control described above, displayed many ambiguities, the application of the algorithm and the software allowed data to be encoded as unknown in order for us to impute the data either via the algorithm or through manual checking.

Following some exploratory analysis, we chose to reconstruct the primary architecture of the global phylogenetic network of SARS-CoV-2 variation by using only haplotypes that were present at absolute frequency higher than 10 in the database. We used the three outgroup sequences to root the tree by including them in the network analysis. These specified well-defined clades and excluded the possibility of inferring artefactual branches generated by low-quality sequences. As this reconstruction proved stable even after adding the low-frequency haplotypes, and since various conflicting nomenclatures have been proposed, none of which has been grounded in a reliable rooting of the tree, we used this network to create a new working nomenclature for the phylogeny. For this, we used letter labels established by GISAID/nextstrain (A and B) intercalated with numbers (e.g., A1, A2, B1, B2) and letters (e.g., A1a, A1b, A1a1, A1a1a2) as in the cladistic system established for human mitochondrial DNA [30]. Following this step, we built individual networks for specific clades using all of the data, including singleton haplotypes. The nomenclature had a systematic logic, since we gave priority in the letter or number to the highest frequency subclade in the database for each node, meaning that the most frequent clade was either “a” or “1”, depending on the level.

We discounted the possibility that recombination might adversely affect the analysis. In a highly variable genome, recombination generates distinctive patterns in phylogenetic networks, which we did not observe. Instructively, it has been erroneously claimed several times for human mtDNA [31,32], but the mere detection of small network loops or reticulations involving single nucleotide positions rather than haplotype blocks [33,34] most straightforwardly points to recurrent mutation, not recombination. In fact, reticulation was an extremely minor feature of our phylogeny, much less so than in the demonstrably clonal human mtDNA, occurring primarily around the root.

Recombination has usually been proposed to have affected the virus genome before the jump to humans, perhaps between bat and pangolin viruses [33,35], which would not affect our intraspecific analysis of the human viral genome. However, we point out that even this is in doubt. As Table 1 of Zhou et al. [35] shows, there was no segment of the human SARS-CoV-2 genome closer in sequence to any known pangolin coronavirus genome than to the closest bat coronavirus genome, although the pangolin virus was closer in this segment than other bat virus genomes. In any case, we can be confident that recombination has not affected the phylogeny since the virus began spreading in humans in a way that might impact our phylogeographic analysis.

We colored the networks according to both the geography of the sample (place of collection) for the phylogeographic analysis and date of collection for a chronological analysis. We identified hypothetical founders between the major continents in the networks: Africa, Oceania (mainly Australia), East Asia (mainly China), South Asia, Southwest Asia, Europe, South America, and North America. We considered South Asia and Southwest Asia separately from East Asia for the founder analysis, as the history of these region in terms of SARS-CoV-2 spread seems to derive more directly from Europe than to the ultimate East Asian source region.

For the analyses of intercontinental founders, we considered as candidate founders only those that resulted in at least two samples in the sink continent. By excluding singletons, we confirmed that the founders were haplotypes established within the sink population with at least one hypothetical epidemiological link (i.e., we inferred at least one transmission within the sink population). We analyzed all of the networks phylogeographically and considered all hypothetical intercontinental founders (as direct matches or inferred matches) [36,37]. To establish the direction of the movements, we considered three parameters: (a) the polarity of the tree topology (which depended on the rooting) to establish the origin of the ancestral clade that contains the founder; (b) the observed diversity of the clade in both the hypothetical source and sink, mostly in early detected samples of the clade; (c) the time of detection of the clade in both locations, source, and sink/s. The latter was more dependent than the others on the number of complete genomes sequenced at each location and was mainly used as a criterion only when the first two were inconclusive.

A limitation of the analyses concerned the presence of frequent haplotypes spread across several continents (the “super-spreaders” identified by Gómez-Carballa et al. [17]). In such cases, while it was straightforward to pinpoint the origin of the clade, we could not exclude the possibility that a founder not only moved into a sink population but that this sink population became the source population of the same undifferentiated founder into a second sink population. In addition, the phylogeographic analyses could not distinguish multiple founders of the same haplotype (i.e., haplotypes that were introduced more than once into the sink) but, given that we effectively used the whole genome information from the virus, this would also be true for any genetic analysis applied to the epidemiology of SARS-CoV-2. In practice, this limitation related to a limited number of founders and should not greatly influence the overall results.

## 3. Results

Following the application of quality control criteria on the obtained dataset (Appendix A), we obtained a final dataset of 26,869 SARS-CoV-2 genome sequences for analysis. We built a reduced-median network based on all of the haplotypes that displayed more than 10 occurrences in the dataset: 18,239 sequences (Figure 1). The three outgroup sequences (from viruses extracted from two bats and a pangolin) suggested a more probable root between the two earliest clades detected in China (Figure 1A,B). This made great sense from a phylogeographic perspective.

The defined root lay midway between the short branch separating the two major clades, A and B. It was separated by a single variant (a transition at nucleotide position 8782) from the reference sequence, defining clade A, and a second variant (a transition at nucleotide position 28144), defining clade B (Figure 1A; Appendix A). Both clades A and B were present early on in East Asia, with similar levels of diversity (calculated as ρ [38] in Appendix A), supporting the phylogenetic inference phylogeographically.

The network suggested a second possible root due to a reticulation involving one of these two positions (Appendix A). The alternative root was placed between A and A1. However, it was caused by a variant at position 3037, which was revealed as one of the fastest-evolving sites in the genome (35 occurrences: Appendix A). This made it likely that this site evolved twice in the reticulation (causing long branch attraction with the very long branch separating the viruses of the other species), rather than position 8782. Nevertheless, other positions close to the root also have a relatively high mutation rate (mutating 9 times), which explains why results can differ between studies. These are 8782 for A, which mutated 16 times, and 28,144 for B, which mutated 8 times; as well as 18,060 in haplogroup B1, which defines the root in the ML analysis of Gómez-Carballa et al. [18], although they prefer the base of haplogroup A on phylogeographic grounds.. Nevertheless, our inferred root is additionally solid on phylogeographic principles, given the early appearance of both clades A and B in China. Additionally, although there is the possibility of unsampled variation, neither of the mutations at positions 3037 or 18,060 appeared in the early samples sequenced (Appendix A).

We used this network to establish the working cladistic nomenclature shown in Appendix A. We constructed reduced-median networks for all individual clades, generating a final overall phylogeny containing 20,247 mutational events occurring at 8892 different positions of the genome. We established a mutational spectrum of SARS-CoV-2 (Appendix A). Although it was not obvious that any specific regions had differential mutation rates, some positions had a much higher relative mutation rate than others, which was information essential for further phylogenetic reconstruction and could be of interest for functional studies. Appendix A displays the most common mutations in the tree, with the mutation 11,083T showing by far the greatest relative mutation rate.

We show the phylogenies for all the clades in the tree in Appendix A. We used these to identify phylogeographic signals for intercontinental haplotype exchanges, focusing on broad phylogeographic inferences in order to account for sampling bias. The main criterion for understanding the direction of movements was the polarity of the tree structure. It was likely that this structure was captured more accurately by broad continental sampling against the one in individual countries, which could be quite deficient or even non-existent. We identified 301 potential founders between continents, which are listed in Appendix A.

Haplogroup B showed less global spread than its sister clade A. It emerged in China, where the earlier cases were detected and already displayed great diversity in terms of derived branches (Appendix A) before spreading southwards into Southeast Asia. Two subclades then appeared in Europe, B2 (first detected on 25th February in France) and B4 (first detected on 1st March in Spain). Two East Asian subclades, B1 and B3, nested further subclades that appeared in North America, B1a (first sequence dating to 20th February) and B3a (first detected 21st January). The former was seen at a high frequency in the North American dataset, suggesting a major expansion of this subclade. The appearance of subclades of B in Australia seemed to be mainly from North American and European sources, rather than directly from East Asia.

Clade A also had a likely origin in mainland China, as was the case for haplogroup B. Not only were early cases detected but substantial diversity was already present (Appendix A). This diversity included haplogroups A2 and A7. A few subclades that sprung directly from the root of A emerged in both North America (A6 and A8, with first detections at 28th February and 16th March respectively) and Europe (A3 and A4, with first detections at 25th February in Germany and 5th February in England respectively), but some European subclades also emerged nested within East Asian subclades of A (both A1 and A2). The root haplotype of A1 was present in Europe (in Germany) at very low frequency and it seems likely that the extremely common and widespread subclade A1a emerged from A1 in situ within Europe. The earliest sequence of this clade (possibly the “patient zero” for this major expansion) was labelled as having been collected in Italy on 20/02/2020. A1a had a dramatic evolution within Europe, with an extreme star-like distribution giving rise to two subclades, A1a2 and A1a1a, with an equally massive expansion, the former also undergoing a huge radiation in North America as well.

We established the cumulative distribution of the number of founders in each continent/region according to the source of each founder. Again, we focused on the numbers of founders and not their impact (frequency) in the sink population. While the phylogeographic signal could be captured well with even a low sample size, the frequency depended much more on the number of sequences carried out for each country (Appendix A). The results are shown in Figure 2. We used the time of first detection of each founder (Appendix A) minus five days as a proxy for the time of arrival. It is known that the incubation period until symptoms/detection of COVID-19 can be as high as a 11 days, with an average of around five days [39].

The graphics show that, from the end of February and early March, Europe was the source for the vast majority of new founder sequences detected globally (Appendix A) with founders detected in Europe (Figure 2C), North America (Figure 2D), Oceania (Figure 2E) and South Asia (Figure 2G). Transmissions associated with the drastically-expanding A1a and its subclades represented by far the majority of sequenced SARS-CoV-2 genomes in most continents, including Africa (83%) (Figure 2A), North America (68%) (Figure 2D), South America (87%) (Figure 2F), and Oceania (mainly Australia; 60%) (Figure 2E). As mentioned above, A1a was first detected in Italy on 20th February, 2020 (Appendix A). By contrast, China was notably absent from the root of this expansion. Strikingly, however, most new cases in East Asia during this period (with cases dating to between 2nd March and 16th April) belonged to this subclade (Figure 2B), pointing to back-to-East Asia migrations from a European source (in some cases via North America). Nearly half of the sequenced cases were in Southwestern Asia (Figure 2H) and South Asia (Figure 2G).

Following this steep increase in the number of founders during March, the plots begin to level off during April, suggesting that intercontinental transmissions were being brought under control, and most subsequent transmissions occurred through local transmission and evolution of clades already present.

## 4. Discussion

COVID-19, the disease caused by coronavirus SARS-CoV-2, was not the first coronavirus infection to jump species into humans in the last 20 years [40], but it has had by far the highest rate of infection, spreading rapidly throughout the world [10]. The effort to understand and control the pandemic has generated a huge dataset of viral genomes isolated from infected individuals across the globe and at different moments of the pandemic event. Here, we carried out by far the largest phylogeographic analysis to date of SARS-CoV-2 genomes, including almost 27,000 viral sequences. We exploited some similarities to mitochondrial DNA of the viral genome (in number of variants, patterns of variation, and, at the time, in numbers of available samples) to make use of tools successfully developed over the past three decades to analyze intraspecific mitochondrial DNA genomes both phylogenetically and phylogeographically, in order to infer patterns of dispersal and founder effects [19]. This enables us to explore the phylogeographic signals in SARS-CoV-2 genomes in much greater depth than has been possible using more traditional phylogenetic approaches that were designed for interspecific datasets (e.g., van Dorp et al. [41]). Alternative attempts to circumvent the difficulties of tackling a large intraspecific dataset appear to have been less successful, since the branching structure they propose fails to match the one we demonstrate here [42].

Here, we extended the initial work of Forster et al. [16] and Gómez-Carballa et al. [17] by applying the analysis to more than two orders of magnitude more sequences than the earlier study and almost an order of magnitude more than the latter. This allowed us to focus on intercontinental founder events, which is crucial evidence for understanding not only the spread of the pandemic but the effectiveness of public health measures. Unlike the earlier studies, our analysis was founded on reduced-median networks, providing a more rigorous phylogenetic basis for the topology of the viral tree of descent, including the identification of the root of the phylogeny, when the virus first made the jump from an unknown species to humans.

This was most likely horseshoe bats [43,44], as for SARS-CoV-1 and MERS-CoV [4,45], although, as the sequence identity to the closest bat sequence was only 96%, this remains controversial [41]. Host switching is common in bat coronaviruses in China and although bats most likely form the main reservoir, pangolins represent a second possible source [8,46,47]. However, sequence identity with the pangolin coronavirus is substantially lower (below 90%), and if it made a contribution to the genome via recombination, prior to the introduction into humans, it could only have been within a short segment of the receptor binding domain of the Spike protein thought to mediate binding to the human ACE2 protein [35,48]. However, as we noted above, since even this segment of the SARS-CoV-2 genome is closer to the most similar bat viral genome, even this seems unlikely. This possibility does not invalidate the use of bat viral genomes as outgroups.

Much of the ensuing analysis depends on correctly locating the root of the phylogeny. The root we identified differed substantially from those detected by both Forster et al. [16] and Gómez-Carballa et al. [17,18]. Because our inferred root was equidistant from the two internal nodes that were sampled at the earliest date in China, which generated similar diversity, we regarded it as much more plausible than the various alternatives that have been proposed. By focusing initially on multiple sequence haplotypes, to avoid being misled by sequencing ambiguities and artefacts, we generated a complete phylogenetic network for SARS-CoV-2 genomes and a robust outline for the phylogenetic tree. In fact, the complete reduced-median network itself is extremely tree-like; there were few reticulations indicating ambiguities in the topology.

We conclude that we have an accurate and rather comprehensive sketch of the first six months of the evolution and dispersal of the virus, with the caveat that sampling was not carried out evenly around the world, up until mid-May 2020. We largely restricted our inferences to intercontinental rather than intracontinental dispersal of the virus, in order not to be misled by sampling biases from one country to another. In particular, the majority of European samples were from the United Kingdom, with countries such as Italy—which was at the root of the European outbreak—poorly represented. It is also true that there was considerable sampling bias from one continent to another, with Europe and North America much more heavily represented than Asia, Oceania, or Africa. However to some extent this also reflected the severity of the pandemic in these regions during the period in question, and the sheer scale of the present dataset means that even the less well-represented continents have a large number of samples.

The strong phylogeographic patterns in the data and their concordance with the time of sampling suggest that the global patterns were well-represented in our networks. For example, despite relatively few samples from China, they were concentrated both in the earliest phases of the outbreak (December 2019 to January 2020) and close to (and either side of) the root of the network, and few of the expansion lineages seen in other continents have been sampled in China. In contrast to van Dorp et al. concluding that “Everything is everywhere” [41] (replaying the conclusion of early analyses of human mtDNA, before the use of intraspecific network analysis: e.g., Pult et al. [49]), we found that SARS-CoV-2 lineages tended to expand in a continent-specific manner, with well-defined major jumps from one continent to another (Appendix A shows the major founders within the branching structure, of which there were fewer than 40). This strongly suggests a clear phylogeographic signal rather than artefacts of sampling bias, and that whilst we must inevitably miss many minor founders, the overall picture is very likely to be correct.

We found that not only was there a large number of intercontinental founder events over this period—at least 300, including all minor founders—but, very strikingly, the vast majority of them originated not in Asia but in Europe. This held true for founders arriving in all regions of the world. Although a small minority of cases in North America originated in East Asia, the great majority spread from Europe. In North America, in fact, cases from Europe continued to arise well into April (and indeed cases from North America continued to spread into Europe over the same period) (Figure 2C,D). Strikingly, many new cases in East Asia also had a European source (Figure 2B).

These results were consistent with the very early dates for the first cases in Europe—in Germany, Italy and France—dating to the end of January 2020, in comparison with other continents, and even other parts of Asia outside China. There were few founders originating in China, and of these just one (A1a) underwent a massive expansion in Europe and beyond. Whilst the source of COVID-19 was clearly in China, the spread of the global pandemic during the first six months was largely fueled by the expansion in Europe.

Why this particular subclade underwent such a huge expansion is not obvious. The single mutation defining subclade A1a is a transition in the RNA polymerase gene and it is not clear that this would be likely to provide the virus with enhanced infectivity. Gómez-Carballa et al. [18] found few signals of positive selection in the virus genome. However, one of the three variants that defines subclade A1, mutation 23043 in the SARS-CoV-2 Spike protein, was correlated with higher virus load in patients, while not necessarily correlated with increased disease severity [50]. Thus, it is possible that this mutation was associated with greater infectivity and fueled the spread of the virus. Subclades within A1a account for most of the global dispersal outside of China.

A potential implication of this analysis concerns our understanding of the effectiveness of the various travel bans issued in different countries towards the reduction of intercontinental transmissions. Early studies suggested that travel restriction had limited impact on the spread of the pandemic [51,52]. The US, for example, banned foreign nationals from (or who had visited) China on 31 January, but continued trading activities and allowed US citizens to continue to travel to and from China. Founders continued to arrive until March, although on a far lower scale than the subsequent spread from Europe.

On 11 March 2020, the US extended travel restrictions to foreign nationals who had visited the EU Schengen Area countries within the previous two weeks, with the UK and Ireland being included five days later and a recommendation against non-essential travel for US citizens from 19 March 2020. Clearly, the number of founders from Europe did decrease after that period. We considered an average incubation period of around five days to account for the possibility that some founders might be detected later by chance. This showed a total of 16 founders detected in the month after 11 March, against 46 in the month before (about a third). It is clear, however, that the travel ban on foreign nationals, but not on US citizens, did not stop the entrance of new founders in the US. Furthermore, new founders were detected in April, suggesting that the measures did slow down the spread but remained less than fully effective in stopping it.

The pattern was similar with Australia. Australia initially showed the highest number of founders of any country (a total of 82). However, with the establishment of strict controls on the entry of foreign nationals on 20 March and a two-week quarantine instituted from 28 March, new founders fell drastically from 65 in the month before to 13 in the month following (down to a fifth). This was at a time when Australia was headed towards winter, which might be thought to exacerbate the spread of the virus. Although the measure was not fully effective, it seems to have been more so than in the US. In fact, travelling from China had been previously banned from 1st February and only one founder was detected in Australia on the 21st February from a possible East Asian source.

Much more dramatic results were evident for East Asia. China established much stricter travel restrictions on entry to virtually all foreign nationals by 27 March. By 1st April, most East Asian countries had implemented strict travel restrictions. The data suggest that the arrival of new founders ceased entirely before the end of March. However, when the measures were effective overall, it was difficult to disentangle the effects of travel restrictions from those of very strict lockdowns.

For Africa, South America, South Asia, and Southwestern Asia, the data are much less robust. For all regions, it seems that strict travel restrictions may have substantially reduced the number of founders, although they did not prevent them altogether. For example, India introduced travel bans by 18th March and the number of founders in the month before and after corresponded to 10 and 3, respectively. Once again, however, it is difficult to disentangle the positive results from the effect of other measures, since Europe, the region that was largely seeding the pandemic through March and early April, was almost entirely under lockdown from the second half of March.

For Europe, although the EU restricted the arrival of non-EU nationals after 17th March, the border-free Schengen Area between most EU countries remained open. Travel restrictions within the continent were introduced by some countries to a variable extent at varying times, with various piecemeal attempts to implement self-quarantine regimes. For example, the UK quarantine measures and travel restrictions were only brought in in June 2020. Climate may also have been a confounding factor and it is not entirely clear how the number of founders correlates with the spread of the pandemic.

However, by March, the virus was already widespread throughout Europe, which had become the epicenter of the pandemic. Most of the evolution of SARS-CoV-2 in Europe was the result of local spread within the continent, reflecting the fact that the borders that remained open across most of Europe. Even so, by the end of March and in April, new founders were still being detected (14 founders), all from a North American source, suggesting that even intercontinental travel restrictions were less effective between Europe and the US. Given the massive proliferation that had taken place within Europe, any such permeability remained a potential risk for other parts of the world.

We demonstrated here that the phylogeographic detection and analysis of founder lineages can be a powerful approach with which to track intercontinental movements of a fast-spreading pathogen in fine detail. Not only that, but it has practical value in allowing us to evaluate the impact of travel restrictions imposed at various times around the world. We highlighted how some were more effective than others in slowing down the spread, but also by unequivocally demonstrating that none were wholly effective in halting the spread. Air travel has clearly been a major driver of the pandemic, and our analysis suggests that stricter travel restrictions appear to be more effective in slowing the spread. Insights from evolutionary analysis can therefore also serve as a learning experience to help manage current and future threats to public health.

## Figures and Tables

**Figure 1 microorganisms-08-01678-f001:**
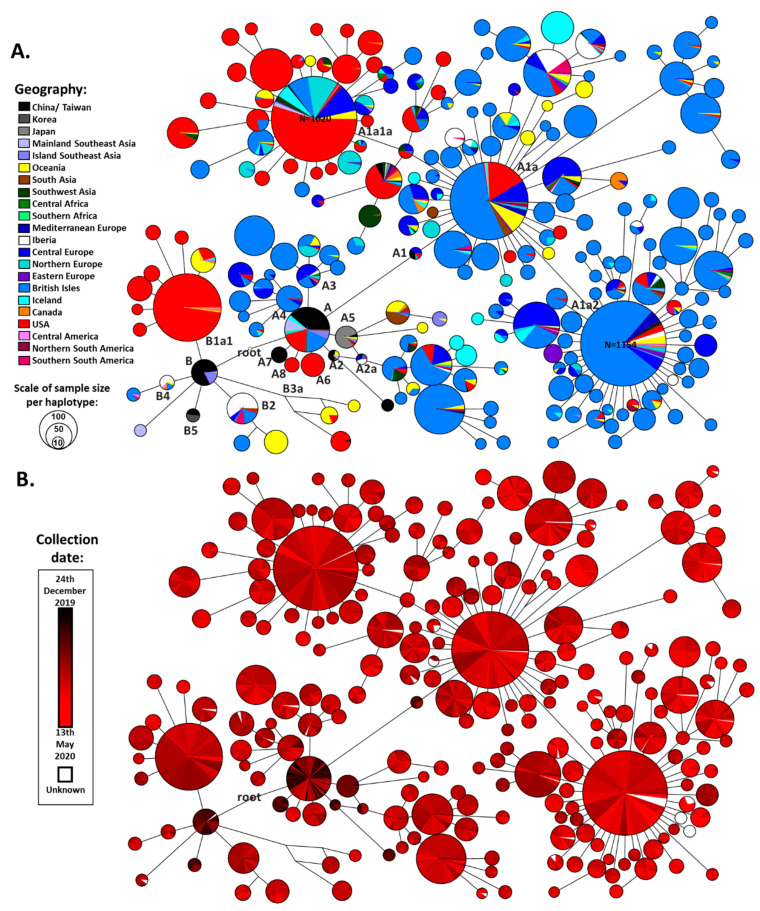
Reduced-median network of the most common haplotypes in the SARS-CoV-2 genomic database. Networks are colored according to geography (**A**) and the time of collection (**B**). Size of the circles corresponds to the number of samples with the same haplotype and a scale is provided. Two haplotypes have more samples than the limit of the software and the number of samples is provided. Links correspond to inferred mutations between haplotypes. Given the complexity of the network, the length of the links is not fully additive (i.e., exactly scaled to the number of mutations). A defined nomenclature for the nodes of this network is presented in Appendix A, Appendix A displays the mutations between haplotypes, and branch lengths are scaled to the number of mutations in the networks in the individual networks (Appendix A).

**Figure 2 microorganisms-08-01678-f002:**
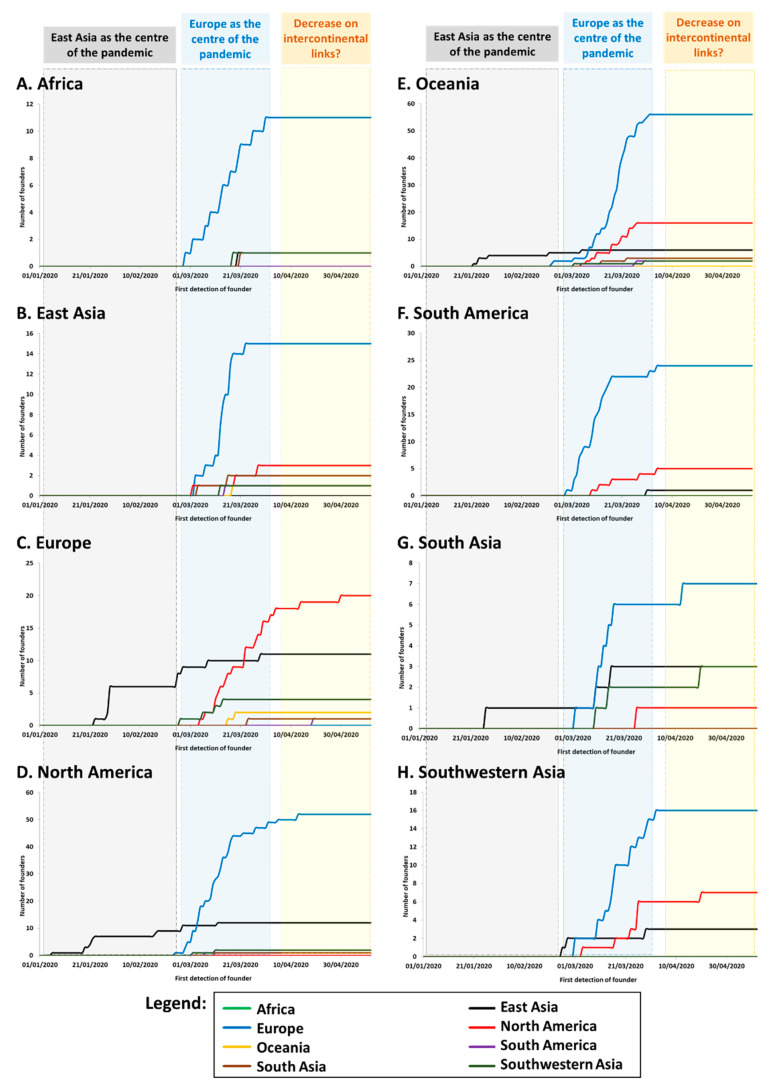
Accumulative distribution of the number of founders in each continent/region from different sources for: (**A**). Africa, (**B**). East Asia, (**C**). Europe, (**D**). North America, (**E**). Oceania, (**F**). South America, (**G**). South Asia, and (**H**). Southwestern Asia.

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
