# Peer review of "Phylogeography of 27,000 SARS-CoV-2 Genomes: Europe as the Major Source of the COVID-19 Pandemic"

_microorganisms, 2020, doi:10.3390/microorganisms8111678_

Round 1

Reviewer 1 Report

In this manuscript, Rito et al collected 27,000 SARS-CoV-2 genomes and performed a comprehensive phylogeographic analysis. The authors estimated that a viral strain in Europe became a main actor of the subsequent spread worldwide although China was an origin of the pandemic. This lineage accounted for the great majority of cases detected globally since the global pandemic was mainly promoted by its expansion across and out of Europe.

Overall, the manuscript presents interesting data on the phylogeographic approach of SARS-CoV-2 pandemic. However there is several points to be addressed before publication. In particular, the limitation of the study with regional bias in the dataset should be explained. My specific comments are listed below.

1)  There is a prominent regional deviation between groups of datasets analyzed in this study. For example, in Figure 1A, majority of cases in Europe are derived from the British Isles, while there is very little analysis in other European countries (such as Italy) and Asian countries. The similar deviation was also seen in Figure 2. Due to the limited data, the spread from the North American or Asian region to each area might have been underestimated. The authors should address the important point.

2) Figure 1A: The most strains are derived from A1a. It this because A1a is the large number of strains in the data set, or was the mutation A1a to A1a1a or A1a2 mutation affected viral growth? Authors should discuss the effect of the mutation on viral replication.

3) Figure 2: The strains originated from Europe exhibited the same mutation in each area it migrated? Is there regional variation in this mutation?

4) The author indicates that Europe is the origin of the influx of new virus strains into each region. There is no description in the paper on the subsequent transition of the detected virus strain. Therefore, it would be an overinterpretation that strains from Europe have spread all over the world by data in Fig 1a and 1b. 

5) As a factor of the expansion of the virus strain of the Europe to the world, the authors raise the mobilization of people from Europe. It would be better if the authors address viral factors or cultural customs in each area.

6) There is no scale indicator of reduced-median network in Figure 1a.

7) A great deal of the material covered in the current manuscript has been addressed in prior publications. The authors should make sure the novelty of this study as compared with literature.

Author Response

1) There is a prominent regional deviation between groups of datasets analyzed in this study. For example, in Figure 1A, majority of cases in Europe are derived from the British Isles, while there is very little analysis in other European countries (such as Italy) and Asian countries. The similar deviation was also seen in Figure 2. Due to the limited data, the spread from the North American or Asian region to each area might have been underestimated. The authors should address the important point.

Response: We thank the reviewer for the comment. We agree that the different sample sizes are a matter of concern. However, detection of founders and probably sources are not based on frequencies but instead on phylogenetic tree structure that can be captured based on a lower amount of samples. We added to the discussion (lines 325-343):

“We largely restrict our inferences to intercontinental, rather than intracontinental, dispersal of the virus, in order not to be misled by sampling biases from one country to another, which are very substantial. In particular, the majority of European samples are from the United Kingdom, with countries such as Italy – which was at the root of the European outbreak – poorly represented. It is also true that there is considerable sampling bias in sampling from one continent to another, with Europe and North America much more heavily represented than Asia, Oceania or Africa, but to some extent this also reflects the severity of the pandemic in these regions during the period in question, and the sheer scale of the present dataset means that even the less well-represented continents have a large number of samples.

“The strong phylogeographic patterns in the data and their concordance with the time of sampling do indeed suggest that the global patterns are well-represented in our networks. For example, despite relatively few samples from China, they are concentrated both in the earliest phases of the outbreak (December 2019 to January 2020) and close to (and either side of) the root of the network, and few of the expansion lineages seen in other continents have been sampled in China. In contrast to van Dorp et al.’s conclusion [41] of “Everything is everywhere” (replaying the conclusion of early analyses of human mtDNA, before the use of intraspecific network analysis: e.g. [49]), we find that SARS-CoV-2 lineages tend to expand in a continent-specific manner, with well-defined major jumps from one continent to another (Figure S1 shows the major founders within the branching structure, of which there are fewer than 40). This strongly suggests a clear phylogeographic signal rather than artefacts of sampling bias, and that whilst we must inevitably miss many minor founders, the overall picture is very likely to be correct.”

2) Figure 1A: The most strains are derived from A1a. It this because A1a is the large number of strains in the data set, or was the mutation A1a to A1a1a or A1a2 mutation affected viral growth? Authors should discuss the effect of the mutation on viral replication.

Response: A1a contains by far the largest number of strains in the dataset. We added to the discussion the relevance of the three mutations that define A1 and the one that defines A1a (lines 3577-364):

“Why this particular subclade underwent such a huge expansion is not obvious. The single mutation defining subclade A1a is a transition in the RNA polymerase gene and it is not clear that this would be likely to provide the virus with enhanced infectivity. We note that Gómez-Carballa et al. [18] found few signals of positive selection in the virus genome. However, one of the three mutations that defines subclade A1, mutation 23043 in the SARS-CoV-2 Spike protein, has been correlated with higher virus load in patients, while not necessarily correlated with increased disease severity [50], so it is possible that this mutation is indeed associated with greater infectivity and has fueled the spread of the virus. Subclades within A1a do indeed account for most of the global dispersal outside China.”

3) Figure 2: The strains originated from Europe exhibited the same mutation in each area it migrated? Is there regional variation in this mutation?

Response: All the strains exhibited the same set of mutations (defining A1a) and there are specific mutations that define subclades (displayed in detail in the networks of the supplementary material), which define the different founders.

4) The author indicates that Europe is the origin of the influx of new virus strains into each region. There is no description in the paper on the subsequent transition of the detected virus strain. Therefore, it would be an overinterpretation that strains from Europe have spread all over the world by data in Fig 1a and 1b. 

Response: As indicated in the manuscript, lineage A1 occurs in East Asia and Europe and its derived subclade A1a is clearly of European origin, most likely from Italy. That lineage and its further derived subclades became the most representative lineage worldwide. We have clarified this by expanding our description of  the impact of this lineage worldwide ad follows (Table S3 and the text:

“Transmissions associated with the drastically-expanding A1a and its subclades represent by far the majority of sequenced SARS-CoV-2 genomes in most continents, including Africa (83%) (Figure 2A), North America (68%) (Figure 2D), South America (87%) (Figure 2F) and Oceania (mainly Australia; 60%) (Figure 2E). A1a was first detected in Italy on 20th February, 2020 (Table S3).  By contrast, China is notably absent from the root of this expansion.

5) As a factor of the expansion of the virus strain of the Europe to the world, the authors raise the mobilization of people from Europe. It would be better if the authors address viral factors or cultural customs in each area.

Response: We are studying the spread of a pathogen between global areas and factors like cultural customs could of course influence the spread of the virus locally but we felt that is really a topic in itself and beyond what we can do here – although we touch on it in our discussion of the different ways that travel restrictions were implemented in different parts of the world. Regarding virus factors, as we mention above (2), we now discuss the possibility that A1 might contain a mutation that increases virulence and could have helped its spread across the globe.

6) There is no scale indicator of reduced-median network in Figure 1a.

Response: Figure 1 summarises a huge amount of data that leads to difficulties in displaying all possible information, including branch lengths. Thus, in the summary figure they are not always entirely additive, in phylogenetic terms (i.e. exactly proportional to the number of mutations). We have now made this point explicitly in the figure, and for additional clarity we also added a scale on the sample size for each haplotype. We modified the legend to Figure 1 as follows:

“…Size of the circles correspond to the number of samples with the same haplotype and a scale is provided. Two haplotypes have more samples than the limit of the software and the number of samples is provided. Links correspond to inferred mutations between haplotypes. Given the complexity of the network the length of the links is not fully additive (i.e. exactly scaled to the number of mutations) to the number of mutations. A defined nomenclature for the nodes of this network is present in Figure S2, Figure S1 displays the mutations between haplotypes, and branch lengths are scaled to the number of mutations in the networks in the individual networks (Figures S7–S57).”

7) A great deal of the material covered in the current manuscript has been addressed in prior publications. The authors should make sure the novelty of this study as compared with literature.

Response: We have now further emphasized what is new and what was shown before in various new sections of the manuscript. These include the first paragraph of the Discussion, as follows (lines 291-296):

“This enables us to explore the phylogeographic signals in SARS-CoV-2 genomes in much greater depth than has been possible using more traditional phylogenetic approaches that were designed for interspecific datasets (e.g. [41]). Alternative attempts to circumvent the difficulties of tackling a very large intraspecific dataset appear to have been less successful, since the branching structure they propose fails to match the one we demonstrate here [42].”

Reviewer 2 Report

Important findings like these should be published in Microorganisms. The manuscript id very well written and extremely interesting. Suggestions to improve the manuscript are:

  1. What other studies (on other virus genomes) have made use of the same methodology? What are the limitations of these studies compared to yours?
  2. Please provide a little more detail in the figure legends to help orient the reader. 
  3. Are the authors implying that air travel is a major driver of pandemics such as COVID-19? If so, what recommendations are they making for the future? Air travel would be much more feasible to control than other means to slow the spread of pandemics. 

Author Response

1) What other studies (on other virus genomes) have made use of the same methodology? What are the limitations of these studies compared to yours?

Response: The current information on genomics of SARS-CoV-2, collected globally in a short period, displays the ideal characteristics for the current application using median networks. Various papers (mostly preprints) applied this, but using the median-joining algorithm that is easy to implement but very inappropriate for these kinds of data. We now emphasize this further in several passages:

(lines 63-67): “Although a more precise phylogenetic reconstruction is considerably more challenging and labour-intensive, MJ does not provide a reliable short-cut, and at minimum should be combined with a pre-processing step using the reduced-median network algorithm, as recommended by its originators [22]. Other studies currently available only as preprints have also adopted the MJ approach e.g. Song et al. [24].”

(lines 78-80): “The RM algorithm explicitly reconstructs ancestral nodes in the phylogeny as median vectors, of which MJ only achieves a small subset [22,26]. The RM algorithm was explicitly developed for intraspecific human mtDNA variation [26] and as such is highly appropriate because the variation in SARS-CoV-2 genomes is in fact strikingly similar to that in whole mitochondrial genomes, and the number of viral genomes that we analyse (~27,800) is in fact very similar to the entire publicly available whole mitochondrial genome database (PhyloTree) [27].”

2) Please provide a little more detail in the figure legends to help orient the reader. 

Response: We added more detail to figure 1 (see above: referee 1 (6)), which we assume  was the one the referee was referring to.

3) Are the authors implying that air travel is a major driver of pandemics such as COVID-19? If so, what recommendations are they making for the future? Air travel would be much more feasible to control than other means to slow the spread of pandemics.

 Response: We agree, as the virus spread across different continents, air travel must have played an important role. We have now added the following to the discussion on this to make the point explicitly (lines 427-429):

“Air travel has clearly been a major driver of the pandemic, and our analysis suggests that stricter travel restrictions appear to be more effective in slowing the spread.”

Reviewer 3 Report

Review of "Phylogeography of 27,000 SARS-CoV-2 genomes Europe as the major source of the COVID-19 pandemic" by Rito et al.

The authors have studied the growing database of SARS-CoV-2 genomes. This is not the first study that uses more than 20,000 genomes: one from Portugal has been cited, but not yet the recent important one by Song et al. (https://www.biorxiv.org/content/10.1101/2020.08.30.273235v2) introducing 2019nCoVR: this data resource "generates visualization of the spatiotemporal change for each variant and yields historical viral haplotype network maps for the course of the outbreak from all complete and high-quality genomes".
There are also numerous previous studies that presented (partial) networks of SARS-CoV-2 genomes (with different rootings), e.g. Fereira, Akther et al., Bhattacharjee et al., Thakur et al., Ngoc Minh Hien Phan et al., Srivastava et al., Ahmad et al., Farah et al., Kemenesi et al., Gong et al., Fang et al., and Yu et al. All these studies, some of which are preprints, used the median-joining algorithm (MJ) for graphical display. Dogan et al. (doi:10.20944/preprints202005.0332.v2) claimed a Guangdong origin of SARS-CoV-2.
The authors of the paper under consideration have decided to use the reduced median network algorithm (RM) instead of MJ, which indeed offers some advantages, but is not necessarily "a more rigorous phylogenetic basis for the topology of the viral tree of descent", as the authors claim. One can use both, even in parallel. The RM method applied to very large datasets needs a hierarchical design, which makes full sense here in highlighting the initial outbreak of the virus infection: the frequent genomes and those sampled at an early stage would form basal networks, respectively, to which the more peripheral "clades" (shown in the Supplement of the Rito paper) can be attached. Well done.
But do we really know that they are always "clades"? Coronaviruses undergo recombination. How frequent is recombination in SARS-CoV-2? Neither RM nor MJ are suitable for pinpointing recombination. So, this needs extra efforts. Normally, more than just one single strain gets transmitted, so that recombinants could also occasionally bear parts of distant strains within the quasi-species of SARS-CoV-2. Yi (2019 Novel Coronavirus Is Undergoing Active Recombination; 884 • cid 2020:71 (1 August) • CORRESPONDENCE ) and Chaw et al. (Journal of Biomedical Science (2020) 27:73) claim that they found evidence for recombination – can the authors support or refute that? In any case, Zhou et al. concluded in March that the evolutionary pattern is indicative of a complex combination of recombination and natural selection. Thus, one outgroup alone does not suffice, since the spike protein has originated in a second source. In bats, host switching of (beta) coronaviruses is not infrequent (Latinne et al., see doi: 10.1038/s41467-020-17687-3).
At the beginning of their text Rito et al. mention the Huanan live animal and sea-food market of Wuhan as the source of origin of the virus spread. This is rather a popular myth, which is not really conceivable since a bat virus cannot thrive in humans without longer and intense contacts with the same group of humans in order to adapt, say, by recombining with already adapted viruses. A more likely explanation is therefore provided by the "Mojiang" scenario (https://swprs.org/covid-19-virus-origin-the-mojiang-miners-passage-hypothesis/) given by Latham & Wilson in their article (https://www.independentsciencenews.org/commentaries/a-proposed-origin-for-sars-cov-2-and-the-covid-19-pandemic/).
Finally, the paper highlights intercontinental dispersal of the virus strains, which may be of interest in its own right. Even when "the effectiveness of the early travel restrictions implemented by some countries" can be shown to have reduced the number of founder viruses, I have some doubts whether this is relevant to the epidemic situation: Then simply fewer virus strains would have done the job of the many strains.
When discussing the patterns of the global spread of SARS-CoV-2 the authors ignore the issue of seasonal and climatic patterns. Most of Europe and the U.S.A. show a peak in March, since the virus arrived in winter. The epidemic takes 80 days (or a little more in larger countries) irrespective of the measures taken. The tropical pattern already found in Mexico and the southern U.S.A. is somewhat different. In Australia and New Zealand the virus arrived at the wrong time of the year, so that most transmissions came from outside and could not well continue within the country. This has little to do with the measures taken. To shut down borders may somewhat reduce the number of founders crossing borders, but will have little impact on the development of an epidemic as long as there are viruses thriving within the country.

Author Response

1) The authors have studied the growing database of SARS-CoV-2 genomes. This is not the first study that uses more than 20,000 genomes: one from Portugal has been cited, but not yet the recent important one by Song et al. (https://www.biorxiv.org/content/10.1101/2020.08.30.273235v2) introducing 2019nCoVR: this data resource "generates visualization of the spatiotemporal change for each variant and yields historical viral haplotype network maps for the course of the outbreak from all complete and high-quality genomes".
There are also numerous previous studies that presented (partial) networks of SARS-CoV-2 genomes (with different rootings), e.g. Fereira, Akther et al., Bhattacharjee et al., Thakur et al., Ngoc Minh Hien Phan et al., Srivastava et al., Ahmad et al., Farah et al., Kemenesi et al., Gong et al., Fang et al., and Yu et al. All these studies, some of which are preprints, used the median-joining algorithm (MJ) for graphical display. Dogan et al. (doi:10.20944/preprints202005.0332.v2) claimed a Guangdong origin of SARS-CoV-2.
The authors of the paper under consideration have decided to use the reduced median network algorithm (RM) instead of MJ, which indeed offers some advantages, but is not necessarily "a more rigorous phylogenetic basis for the topology of the viral tree of descent", as the authors claim. One can use both, even in parallel. The RM method applied to very large datasets needs a hierarchical design, which makes full sense here in highlighting the initial outbreak of the virus infection: the frequent genomes and those sampled at an early stage would form basal networks, respectively, to which the more peripheral "clades" (shown in the Supplement of the Rito paper) can be attached. Well done.

Response: We thank the referee for these comments. We have now referenced the Song et al. preprint, in the context of the widespread use of the median-joining algorithm. Most of the other references are also preprints, not peer-reviewed, and we felt it might be excessive to cite very many preprints. Most of these also used the median-joining algorithm. Contrary to what the reviewer says, reduced-median (which one of us helped to develop) is certainly a more rigorous phylogenetic basis for the topology of the viral tree of descent. We added quite substantially to the discussion of this:

(lines 63-67): “Although a more precise phylogenetic reconstruction is considerably more challenging and labour-intensive, MJ does not provide a reliable short-cut, and at minimum should be combined with a pre-processing step using the reduced-median network algorithm, as recommended by its originators [22]. Other studies currently available only as preprints have also adopted the MJ approach, e.g. Song et al. [24].”

 (lines 78-80): “The RM algorithm explicitly reconstructs ancestral nodes in the phylogeny as median vectors, of which MJ only achieves a small subset [22,26]. The RM algorithm was explicitly developed for intraspecific human mtDNA variation [26] and as such is highly appropriate because the variation in SARS-CoV-2 genomes is in fact strikingly similar to that in whole mitochondrial genomes, and the number of viral genomes that we analyse (~27,800) is in fact very similar to the entire publicly available whole mitochondrial genome database (PhyloTree) [27].”

2) But do we really know that they are always "clades"? Coronaviruses undergo recombination. How frequent is recombination in SARS-CoV-2? Neither RM nor MJ are suitable for pinpointing recombination. So, this needs extra efforts. Normally, more than just one single strain gets transmitted, so that recombinants could also occasionally bear parts of distant strains within the quasi-species of SARS-CoV-2. Yi (2019 Novel Coronavirus Is Undergoing Active Recombination; 884 • cid 2020:71 (1 August) • CORRESPONDENCE ) and Chaw et al. (Journal of Biomedical Science (2020) 27:73) claim that they found evidence for recombination – can the authors support or refute that? In any case, Zhou et al. concluded in March that the evolutionary pattern is indicative of a complex combination of recombination and natural selection. Thus, one outgroup alone does not suffice, since the spike protein has originated in a second source. In bats, host switching of (beta) coronaviruses is not infrequent (Latinne et al., see doi: 10.1038/s41467-020-17687-3).

Response: In fact, RM networks can be and have been useful for detecting recombination when there is plentiful variation available with which to pick up the presence of haplotype blocks.  In our analysis, the reticulations we observed through the RM networks were in fact extremely minimal and most likely the result of recurrent mutations, as we now discuss extensively in the text. Regarding the use of a single outgroup, in fact we used three (each phylogenetically different, including one pangolin). We did not find any evidence for recombination. We have now added substantially to the text, including the references the referee mentions, and we thank the referee for raising this interesting and important topic:

(lines 134-149): “We discounted the possibility that has been raised occasionally that recombination might adversely affect the analysis. In a highly variable genome, recombination generates distinctive patterns in phylogenetic networks which we do not observe. Instructively, it has been claimed erroneously several times for human mtDNA [31,32], but the mere detection of small network loops or reticulations involving single nucleotide positions rather than haplotype blocks [33,34] most straightforwardly points to recurrent mutation, not recombination. In fact, reticulation is an extremely minor feature of our phylogeny – indeed, much less so than in the demonstrably clonal human mtDNA – and occurs primarily around the root.

“Recombination has usually been proposed to have affected the virus genome before the jump to humans, perhaps between bat and pangolin viruses [33,35] which would not affect our intraspecific analysis of the human viral genome. However, we would point out that even this is in doubt: as Table 1 of Zhou et al. [35] show, there is no segment of the human SARS-CoV-2 genome that is closer in sequence to any known pangolin coronavirus genome than to the closest bat coronavirus genome, although the pangolin virus is closer in this segment than other bat virus genomes. In any case, we can be confident that recombination has not affected the phylogeny since the virus began spreading in humans, in a way that might impact upon our phylogeographic analysis.”

(lines 306-315): “This was most likely horseshoe bats [43,44], as for SARS-CoV-1 and MERS-CoV [4,45] although, as the sequence identity to the closest bat sequence is only 96%, this remains controversial [41]. Host switching is common in bat coronaviruses in China and although bats most likely form the main reservoir pangolins represent a second possible source [8,46,47]. However, sequence identity with the pangolin coronavirus is substantially lower (below 90%), and if it made a contribution to the genome via recombination, prior to the introduction into humans, it could only have been within a short segment of the receptor binding domain of the Spike protein thought to mediate binding to the human ACE2 protein [35,48]. However, as we noted above, since even this segment of the SARS-CoV-2 genome is closer to the most similar bat viral genome, even this seems unlikely. This possibility therefore does not invalidate the use of bat viral genomes as outgroups.”

3) At the beginning of their text Rito et al. mention the Huanan live animal and sea-food market of Wuhan as the source of origin of the virus spread. This is rather a popular myth, which is not really conceivable since a bat virus cannot thrive in humans without longer and intense contacts with the same group of humans in order to adapt, say, by recombining with already adapted viruses. A more likely explanation is therefore provided by the "Mojiang" scenario (https://swprs.org/covid-19-virus-origin-the-mojiang-miners-passage-hypothesis/) given by Latham & Wilson in their article (https://www.independentsciencenews.org/commentaries/a-proposed-origin-for-sars-cov-2-and-the-covid-19-pandemic/).

Response: This an interesting issue. The Huanan market has been the main hypothesis that appears in the literature, but we now added the citation to the "Mojiang" scenario:

(lines 37-39): “This is thought most likely to have taken place in the Huanan live animal and sea-food market of Wuhan, China, although problems with this scenario have fueled alternatives [2].”

4) Finally, the paper highlights intercontinental dispersal of the virus strains, which may be of interest in its own right. Even when "the effectiveness of the early travel restrictions implemented by some countries" can be shown to have reduced the number of founder viruses, I have some doubts whether this is relevant to the epidemic situation: Then simply fewer virus strains would have done the job of the many strains.

Response: We agree that fewer strains might have the same effect, but we do not think this would contradict the suggestions in our manuscript, which refer to the effectiveness of restrictions on the intercontinental spread of the virus. To avoid misunderstanding, we added (lines 413-414):

“…it is not entirely clear how the number of founders correlates with the spread of the pandemic.”  

5) When discussing the patterns of the global spread of SARS-CoV-2 the authors ignore the issue of seasonal and climatic patterns. Most of Europe and the U.S.A. show a peak in March, since the virus arrived in winter. The epidemic takes 80 days (or a little more in larger countries) irrespective of the measures taken. The tropical pattern already found in Mexico and the southern U.S.A. is somewhat different. In Australia and New Zealand the virus arrived at the wrong time of the year, so that most transmissions came from outside and could not well continue within the country. This has little to do with the measures taken. To shut down borders may somewhat reduce the number of founders crossing borders, but will have little impact on the development of an epidemic as long as there are viruses thriving within the country.

Response: We agree, but the point at which climate affects the spread positively or negatively is far from clear. We also included in the Discussion (lines 390-391):

“…at a time when Australia was heading towards winter, which might be expected to exacerbate the spread of the virus.”

And we also added the following caveat (lines 412):

“Climate may also have been a confounding factor…”